# Exploring the Structural, Electronic, Magnetic, and Transport Properties of 2D Cr, Fe, and Zr Monoborides

**DOI:** 10.3390/ma16145104

**Published:** 2023-07-20

**Authors:** Isabel M. Arias-Camacho, Nevill Gonzalez Szwacki

**Affiliations:** Faculty of Physics, University of Warsaw, Pasteura 5, PL-02093 Warsaw, Poland; isabel.arias@fuw.edu.pl

**Keywords:** transition metal borides, MBenes, energy conversion and storage, 2D magnets

## Abstract

Compared to other 2D materials, MBenes are at an early stage of investigation in terms of both experimental and theoretical approaches. However, their wide range of possible 2D structures leads to novel and challenging properties and consequent applications. From all the possible stoichiometries, we performed a theoretical study of orthorhombic and hexagonal M${}_{2}$B${}_{2}$ MBenes within the framework of density functional theory. We found that both symmetries of Cr${}_{2}$B${}_{2}$, Fe${}_{2}$B${}_{2}$, and Zr${}_{2}$B${}_{2}$ show metallic behavior and could be grown under certain conditions as they were demonstrated to be dynamically stable. Moreover, the values of the magnetic moment observed, in specific ferromagnetic cases exceeding $2.5\;\mu_{B}$/M${}_{2}$B${}_{2}$, make them suitable as robust 2D magnets. Our findings represent an important step in the understanding of MBenes and open several windows to future research in fields like energy conversion and storage, sensing, catalysis, biochemistry, and nanotechnology, among others.

## 1. Introduction

Transition metal borides (MBs) can be regarded as new efficient earth-abundant materials for energy storage/conversion systems, including metal-ion batteries, metal-air batteries, capacitors, oxygen evolution reactions (OERs), and other electrochemical fields [1]. Single-crystalline ternary transition metal borides (MAB phases, where M are transition metals, A are *p*-block elements, and B is boron) were first reported in 2015 by Ade and Hillebrecht and recently gained attention as promising layered materials [2]. Their two-dimensional counterparts (MBenes) can be obtained from the chemical exfoliation of the MAB phases. It is noted that MBenes possess different stoichiometries and variable modes of 2D layer sandwiching compared to the corresponding MXenes [3].

The MAB phases with M and B with one-to-one stoichiometry are orthorhombic and hexagonal crystals with the chemical formulae of MAB and M${}_{2}$AB${}_{2}$ [3]. Experimentally, orthorhombic MAlB (M = Mo and W), M${}_{2}$AlB${}_{2}$ (M = Cr, Mn, and Fe), and hexagonal Ti${}_{2}$InB${}_{2}$ [4] have already been synthesized. The MAB phases are promising candidates for obtaining new 2D MBenes. A 2D MoB was reported to have been obtained by the partial etching of Mo${}_{2}$AlB${}_{2}$ phases through the deintercalation of Al layers from the ordered stacking faults region [5,6]. Selectively HCl-etching Al layers from Cr${}_{2}$AlB${}_{2}$ yielded bulk-layered CrB nanosheets [7,8]. Removal of the indium layer through the high-temperature dealloying of Ti${}_{2}$InB${}_{2}$ yielded a bulk-layered TiB structure [4]. To date, however, the synthesis of individual single-layer MBenes has not been realized.

The present work concerns computational studies of the structural, energetic, electronic, and transport properties of selected MBene compounds with M = Cr, Fe, and Zr. Some of the investigated 2D structures have been proven to be stable in previous studies [9]. The studied MBenes present either an orthorhombic structure (ortho-MBenes) with Pmma (no. 51) space group symmetry or a hexagonal structure (hex-MBenes) with P6/mmm (no. 191) space group symmetry. In the Pmma structures, each atom is surrounded by six neighbors, and the buckled bilayers are sandwiched between transition metal (TM) layers. On the other hand, in the P6/mmm structures, the honeycomb-type boron layer is sandwiched between two TM layers on both sides, and every TM atom is located above or below the centroid of the honeycomb structure.

The bulk counterparts of our metal monoboride nanosheets are the ferromagnetic α and β modifications of FeB and the nonmagnetic CrB and ZrB compounds, all widely studied, both experimentally and theoretically [10,11,12]. The structure of α-FeB is debatable [12], whereas β-FeB and CrB are orthorhombic crystals with Pnma (no. 62) and Cmcm (no. 63) space group symmetries, respectively. The ZrB solid is rock-salt-structured and crystallizes in a cubic $Fm\bar{3}m$ (no. 225) space group symmetry. The β-FeB and CrB solids exhibit very interesting structures since both enclose boron double-chain (BDC) stripes which are very common motifs of all-boron nanostructures [13,14].

The purpose of this work is to understand the origin of the physical properties of MBene (M = Cr, Fe, and Zr) compounds, as well as to identify features that may affect the transport properties of these compounds.

## 2. Computational Approach

First-principles spin-polarized calculations were performed within the framework of density functional theory (DFT) within the generalized gradient corrected approximation of Perdew–Burke–Ernzerhof (PBE) [15] for the exchange-correlation functional, using projector plane-wave (PAW) pseudopotentials [16] as implemented in the Quantum ESPRESSO (QE) suite of codes [17]. Every unit cell consists of two atoms of boron and two atoms of the TM. To avoid interactions between adjacent MBenes, we considered an empty space of thickness 15 Å along the normal direction. Optimized geometries were reached allowing the unit cell shape, volume, and the ions to relax until the residual forces on the atoms were less than 0.3 meV/Å and the total energy ($E_{tot}$) convergence was set to 10${}^{-5}$ Ry. We expanded the electronic wave functions and charge density in plane-wave basis sets with an energy cutoff of 70 and 700 Ry, respectively, while the Γ-centered *k*-point grid in the Brillouin zone, in the Monkhorst–Pack scheme, was set to 12×12×1 for the geometry optimization and 24×24×1 for the DOS calculations, with Gaussian smearing of 0.02 Ry; these values ensure the accuracy of $E_{tot}$. All the structures were considered to be initially spin-polarized and, in order to determine the magnetic ground states of each, we calculated two magnetic configurations—one ferromagnetic (FM) and one antiferromagnetic (AFM)—to reach the most energetically favorable.

For the structural characterization of the studied systems, an important descriptor we use is the cohesive energy per atom ($E_{coh}$); that is, the difference in energy between $E_{tot}$ of the compound and the sum of the total energies of the isolated atoms,
(1)$$E_{coh}=\left(E\left[M_{2}B_{2}\right]-n_{B}E\left[B\right]-n_{M}E\left[M\right]\right)/\left(n_{B}+n_{M}\right),$$
which means the released energy when a compound dissociates into isolated free atoms, where M represents the TM atom, $E[M_{2}B_{2}]$ is the $E_{tot}$ of the MBene, E[B] and E[M] are the total energies of the isolated atoms (B and TM atoms), and $n_{B}$ and $n_{M}$ are the numbers of boron and TM atoms per unit cell, respectively, directly obtained from the spin-polarized calculations. The phonon-dispersion curves were obtained by means of density functional perturbation theory (DFPT), calculating the dynamical matrices in the linear response approach on a *q*-point grid of 4×4×1. The transport integrals were computed using Boltzmann transport theory and a constant scattering rate model [18]. The charge transfer was obtained by Bader analysis [19] and all the visualizations were performed using the Visualization for Electronic and STructural Analysis (VESTA) software [20].

## 3. Results and Discussion

### 3.1. Structure and Stability

Since boron is electron deficient, it is expected that a mixture with TM atoms will lead to stable structures. As mentioned above, among all the possible MBenes, we focused on those which possess either orthorhombic or hexagonal M${}_{2}$B${}_{2}$ structures, which are shown in Figure 1. After a full structural optimization, we found that the unit cells of ortho-MBenes became almost rectangular with a>b when the TM was Fe or Cr (a/b is 1.005 and 1.013 for chromium and iron, respectively), whereas a<b for ortho-Zr${}_{2}$B${}_{2}$ (a/b=0.94). All the cell parameters are described in Table 1 and compared with other literature reports.

The $E_{coh}$ and phonon dispersion curves are good indicators of the bond strength and dynamical stability of the materials. The dynamical matrix gives us the frequency ω(q) whose square is negative when there exist instabilities for a particular phonon mode with **q** (imaginary frequencies), which means that this mode does not generate the restoring force needed by the lattice vibrations and could take the structure away from the original configuration. The results of our calculations are summarized in Table 2 and in Figure 2.

All our MBenes exhibit large $E_{coh}$ values ranging from 6.222 to 8.087 eV, as shown in Table 2, which means they present strong internal binding and good stability. Moreover, our results are in good agreement with previous works. For instance, Zhang et al. [28] obtained a value of 6.30 eV for ortho-Cr${}_{2}$B${}_{2}$. For the sake of comparison, we also computed the diamond structure of carbon using the same optimization parameters, resulting in a $E_{coh}$ of 7.757 eV, which is a comparable value to that of other theoretical and experimental reports [29]. Interestingly, other studies have noted a dependence of the structure stability with the atomic mass of the TM [22], a fact that is also reproduced in our results, hex-Zr${}_{2}$B${}_{2}$ being the MBene with the highest $E_{coh}$ (8.087 eV). From a structural point of view, Cr${}_{2}$B${}_{2}$ and Fe${}_{2}$B${}_{2}$ prefer to adopt orthorhombic structures, whereas Zr${}_{2}$B${}_{2}$ accommodates better to the hexagonal one. However, the orthorhombic and hexagonal phases are close in energy (within some tens of meV), and, according to recent reports [9,30], ortho-MBenes might transform into hex-MBenes at high temperatures.

On the one hand, calculations of the phonon frequencies reveal that some small imaginary frequencies appear in the surroundings of the Γ point for orthorhombic Cr${}_{2}$B${}_{2}$ and Zr${}_{2}$B${}_{2}$ as shown on the left panel of Figure 2. On the other hand, none of the hexagonal structures have imaginary frequencies as shown on the right panel of Figure 2. Similar dynamical instabilities as for ortho-MBenes have been reported for freestanding 2D structures in previous works [25,31]. These studies have concluded that the out-of-plane acoustic mode, ZA, is responsible for such instabilities, which are against the long-wavelength transversal waves that could be fixed by defects like grain boundaries or ripples [32].

The highest values of the frequencies at the Γ-point are collected in Table 2. All the frequencies are higher than 740 cm${}^{-1}$. Starting with the orthorhombic structures, the highest optical frequency at Γ-point increases with the atomic number of the TM atom; that is, Cr, Fe, and Zr, in that order. These results are very close to those found in the literature [24]. Since the optical frequency is an indicator of the bond strength, Zr${}_{2}$B${}_{2}$ is more stable than Cr${}_{2}$B${}_{2}$ and Fe${}_{2}$B${}_{2}$. The trend is, however, the opposite for hex-mBenes, meaning that the highest frequency at the Γ-point corresponds to the TM atom which has the smallest atomic number, in this case Cr, and decreases for Fe and Zr. We can conclude that the studied MBenes are stable and could be grown experimentally under certain conditions.

### 3.2. Electronic Properties

To understand the electronic behavior of the MBenes involved in our study, we calculated the spin-polarized band structure, the density of states (DOS), and the projected density of states (PDOS) for the studied systems. The results of these calculations are shown in Figure 3 and Figure 4 for ortho-MBenes and hex-MBenes, respectively. Looking at the band structure and DOS, we may conclude that there are no band gaps between the valence band (VB) and the conduction band (CB) for any of the studied structures, which means that all the systems are metallic with partially occupied bands crossing the Fermi level (for the majority and the minority spin channels). This metallic character of the pristine MBenes has also been reported in other works [33]. In all cases, the *p* orbitals of boron are deep in energy (ranging from −8 to −2 eV approximately) and, in the orthorhombic structures, partially hybridize with the *d* orbitals of the TM. Near the Fermi level, the PDOS for both symmetries of Cr${}_{2}$B${}_{2}$ and Fe${}_{2}$B${}_{2}$ is dominated by the *d* orbitals of Cr and Fe, respectively (see the right panels of (a) and (b) in Figure 3 and Figure 4). Whereas, hybridization between the *p* and *d* orbitals of Zr occurs at the Fermi level of Zr${}_{2}$B${}_{2}$ (see the right panels of (c) in Figure 3 and Figure 4). The contribution of the majority states at the Fermi level of ortho-Fe${}_{2}$B${}_{2}$ is very small, whereas an equal contribution of the minority and majority states is observed in hex-Fe${}_{2}$B${}_{2}$ and also in both symmetries of Zr${}_{2}$B${}_{2}$. Finally, the contribution of the majority states is larger than that of the minority states at the Fermi level for Cr${}_{2}$B${}_{2}$.

According to our Bader analysis, the charge transfer, Δq, always occurs from the TMs to the boron atoms. This is shown in Table 2 where we present the values of Δq for all the studied cases. The obtained values are in agreement with other reports (e.g., Δq=−0.34e for ortho-Fe${}_{2}$B${}_{2}$, as reported in ref. [26]). In general, considering both the Pmma and P6/mmm structures, the largest charge transfer occurs between Zr and B, whereas Fe is the TM for which the charge transfer to B is the smallest one.

The conductivity results are shown in Figure 5. They reveal anisotropy of the conductivity tensor for the ortho-MBenes, especially for the case of ortho-Zr${}_{2}$B${}_{2}$ for which the direction perpendicular to the BDC is clearly the preferred one, whereas for ortho-Fe${}_{2}$B${}_{2}$, the conductivity is higher along the BDC. On the other hand, the hex-MBenes are isotropic with respect to conductivity. It is also worth highlighting that the orthorhombic Fe${}_{2}$B${}_{2}$ and Zr${}_{2}$B${}_{2}$ present the highest conductivity values among all the studied cases.

### 3.3. Magnetic Properties

The origin of magnetism in these types of materials arises from the *d* orbitals of the TM atoms. Both FM and AFM configurations have been suggested to determine the ground state. The results of our calculations are summarized in Table 3. Among all the considered structures, only the hex-Fe${}_{2}$B${}_{2}$ resulted in an AFM ground state since the $E_{tot}$ value of the FM state is 46.84 meV higher in energy. On the other hand, ortho-Fe${}_{2}$B${}_{2}$ exhibited an FM ground state in accordance with previous reports [34]. However, a more detailed analysis, that also included the next nearest neighbors and was performed using DFT combined with the Monte Carlo method, revealed that ortho-Fe${}_{2}$B${}_{2}$ actually has a stable columnar AFM ground state [26]. In the particular case of Cr${}_{2}$B${}_{2}$, both structures result in an FM arrangement of the magnetic moments. In general, the total energy difference, $\Delta E_{\mathrm{FM}-\mathrm{AFM}}$, between the FM and AFM configurations is always higher in absolute value for the orthorhombic structures (−104.38 and −108.12 eV for Cr${}_{2}$B${}_{2}$ and Fe${}_{2}$B${}_{2}$, respectively,) than for the hexagonal ones (−0.29 and 46.84 eV for Cr${}_{2}$B${}_{2}$ and Fe${}_{2}$B${}_{2}$, respectively) indicating that, in the latter case, the magnetic ordering may not be preserved at room temperature. Both orthorhombic Cr${}_{2}$B${}_{2}$ and Fe${}_{2}$B${}_{2}$ structures show an FM ground state with a magnetic moment over 2.5 $\mu_{B}$ per unit cell, a suitable behavior for robust 2D magnets. Interestingly, the boron atoms are also slightly polarized for those cases for which the TM–boron distances ($d_{M-B}$) are the shortest (see Table 1). Finally, Zr${}_{2}$B${}_{2}$ MBenes exhibit non-magnetic behavior.

The energy difference $\Delta E_{\mathrm{FM}-\mathrm{AFM}}=E_{\mathrm{FM}}-E_{\mathrm{AFM}}$ can be used to evaluate the exchange interaction, $J_{\mathrm{NN}}$, between TM atoms at the nearest neighbor (NN) positions. The exchange energy for a system of interacting atomic moments, $S_{i}$, can be described by the Heisenberg model:(2)$$E_{tot}=E_{0}-\frac{1}{2}\sum_{i\neq j}J_{ij}S_{i}\cdot S_{j},$$
where $E_{0}$ is the total energy excluding spin–spin interactions and in our case $S_{i}=S_{j}=S$. For ferromagnetically or antiferromagnetically coupled TM ions at NN positions $-2J_{\mathrm{NN}}S^{2}=\Delta E_{\mathrm{FM}-\mathrm{AFM}}$. The critical temperature (Curie or Néel temperature), $T_{c}$, in the mean field approximation (MFA) can be estimated from
(3)$$T_{c}≃\frac{2}{3}k_{B}\cdot \left(J_{\mathrm{NN}}S^{2}\right)=\frac{1}{3}k_{B}\cdot \left|\Delta E_{\mathrm{FM}-\mathrm{AFM}}\right|.$$

The $T_{c}$ values are collected in Table 3. As can be seen from the table, we obtain a considerably large value of $T_{c}=418$ K for ortho-Fe${}_{2}$B${}_{2}$. However, as mentioned above, a more accurate investigation [26] predicts an AFM ground state with $T_{c}=115$ K. Interestingly, our calculations predict an AFM ground state for hex-Fe${}_{2}$B${}_{2}$ with $T_{c}=181$ K.

## 4. Summary and Conclusions

In summary, we carried out a comparison between the structural, electronic, magnetic, and transport properties for orthorhombic and hexagonal phases of a selected group of MBenes, Cr${}_{2}$B${}_{2}$, Fe${}_{2}$B${}_{2}$ and Zr${}_{2}$B${}_{2}$, that are usually considered separately in the literature. Although there are several theoretical reports that have predicted the stability of the studied MBenes, to our knowledge, hex-Fe${}_{2}$B${}_{2}$ is shown for the first time in this work to be dynamically stable. Experimentally the most studied MBenes are those composed of early TMs, whereas those composed of late TMs (like iron) remain to be synthesized.

We observe that, from an energetic point of view, the larger the atomic weight is, the higher the $E_{coh}$ is, which means that Zr${}_{2}$B${}_{2}$ possesses the strongest bonds. However, despite the small difference between the $E_{coh}$ values of the orthorhombic and hexagonal structures of each MBene, both phases can exist theoretically. This assumption is reinforced by the calculation of the corresponding phonon dispersion plots that predict stable behavior of all the MBenes. Furthermore, the metallic character of our pristine MBenes makes them efficient materials for charge transport, and therefore are competitive 2D materials for electronic, sensing, or electrocatalytic purposes. We predicted that, for both symmetries of Cr${}_{2}$B${}_{2}$, the contribution of the majority spin states would be larger than that of the minority spin states at the Fermi level, leading to an FM ground state and opening the possibility of their use in information magnetic storage.

This understanding of the properties of the studied materials, together with the acknowledgment that, within the same framework, both orthorhombic and hexagonal phases are feasible and different in their properties, creates the possibility of going further in our research to consider them as potential candidates for sensing and catalytic processes. Although there exists some parallelism with MXenes, MBenes are emerging 2D materials that are expected to have great development potential in the future, due to their diverse stoichiometries and, as a consequence, structural differences and new challenging physical, chemical, and biological properties. The biggest difference for MBenes is that some of them can potentially exist in both orthorhombic and hexagonal phases.

## Figures and Tables

**Figure 1 materials-16-05104-f001:**
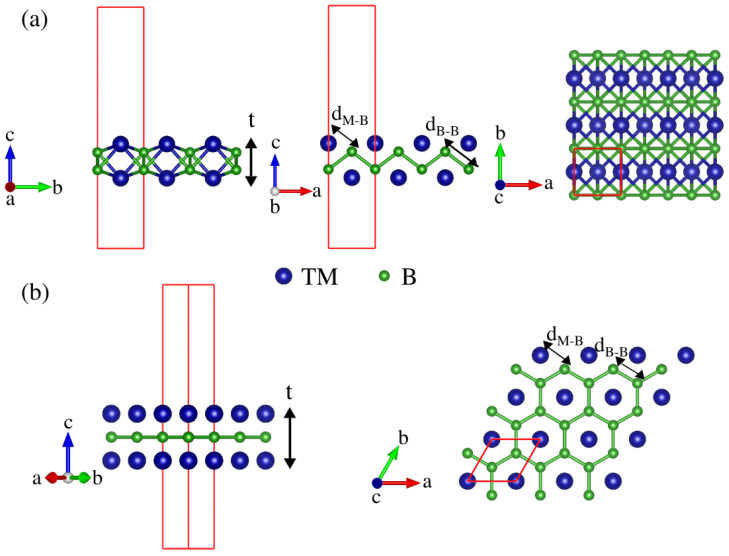
Side (**left**) and top (**right**) views of the (**a**) ortho-MBene and (**b**) hex-MBene structures of M${}_{2}$B${}_{2}$ corresponding to Pmma and P6/mmm symmetries, respectively. The unit cells used in the calculations are shown in red.

**Figure 2 materials-16-05104-f002:**
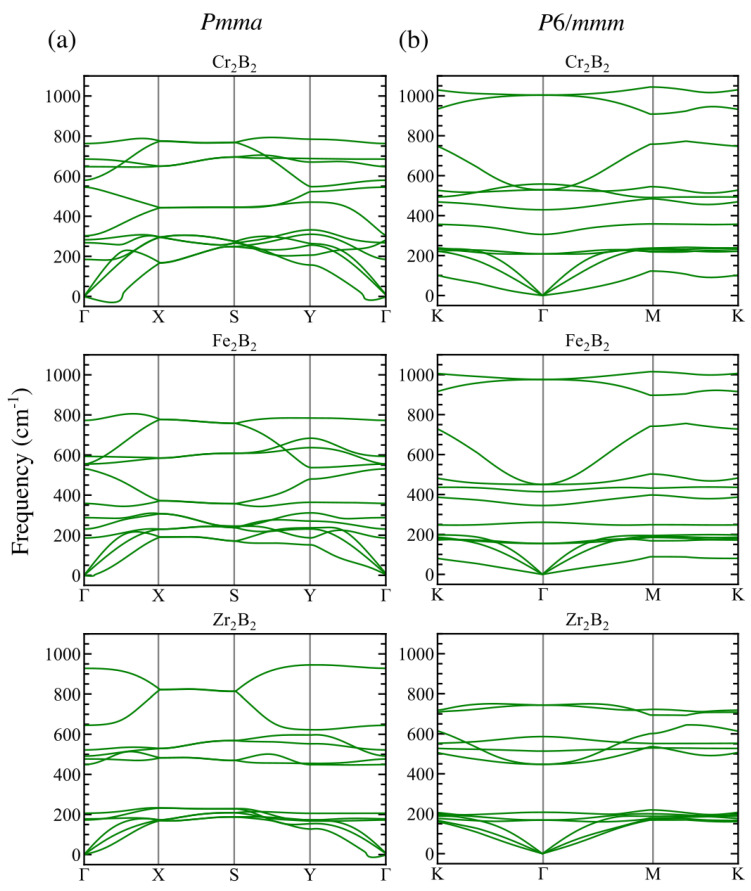
Phonon dispersion relations for the (**a**) Pmma and (**b**) P6/mmm structures of Cr${}_{2}$B${}_{2}$, Fe${}_{2}$B${}_{2}$, and Zr${}_{2}$B${}_{2}$.

**Figure 3 materials-16-05104-f003:**
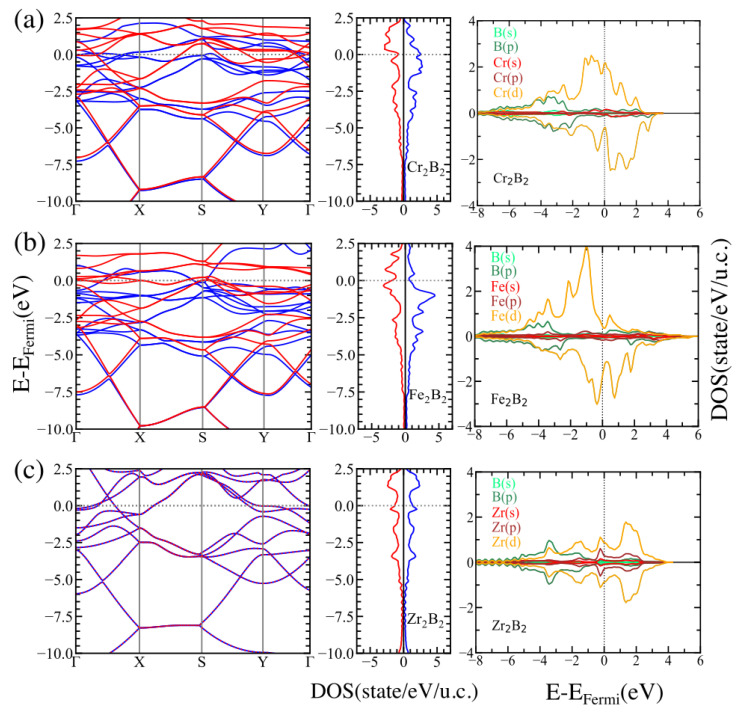
Electronic band structure (**left**) and DOS (**center**) for the majority (blue) and minority (red) spin states, and orbital-resolved PDOS (**right**) for the Pmma structures of (**a**) Cr${}_{2}$B${}_{2}$, (**b**) Fe${}_{2}$B${}_{2}$, and (**c**) Zr${}_{2}$B${}_{2}$.

**Figure 4 materials-16-05104-f004:**
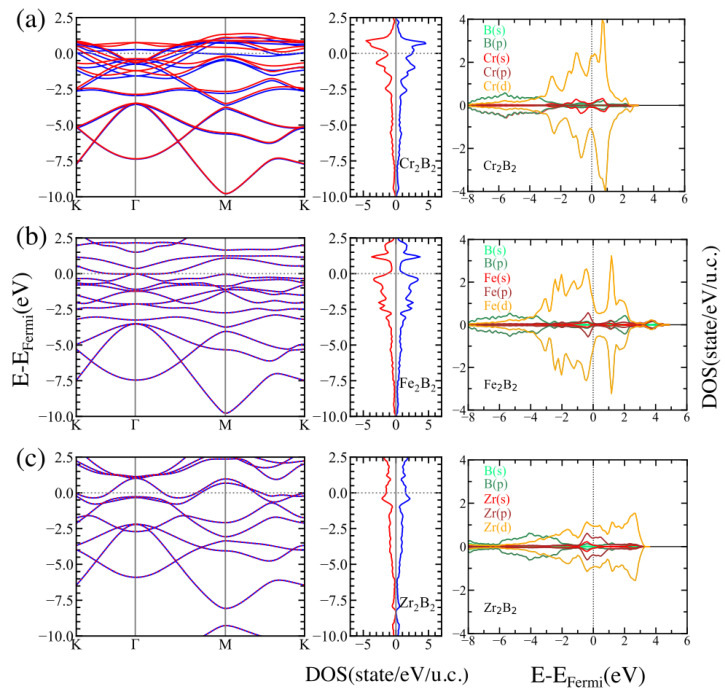
Electronic band structure and DOS for the majority (blue) and minority (red) spins, and orbital-resolved projected density of states (PDOS) for the P6/mmm structures of (**a**) Cr${}_{2}$B${}_{2}$, (**b**) Fe${}_{2}$B${}_{2}$, and (**c**) Zr${}_{2}$B${}_{2}$.

**Figure 5 materials-16-05104-f005:**
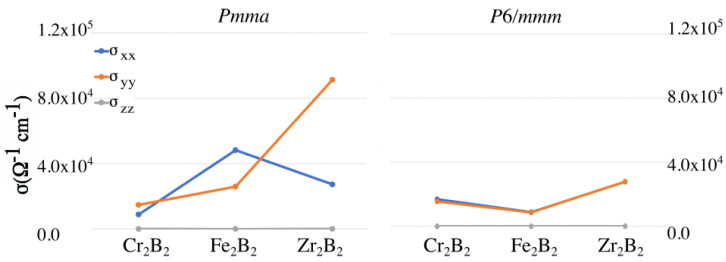
Components of the conductivity tensor for the Pmma (**left**) and P6/mmm (**right**) structures of Cr${}_{2}$B${}_{2}$, Fe${}_{2}$B${}_{2}$, and Zr${}_{2}$B${}_{2}$.

**Table 1 materials-16-05104-t001:** Lattice parameters (*a* and *b*), boron–boron distance ($d_{B-B}$), TM–boron distance ($d_{M-B}$), and thickness of the 2D structure (*t*) for Cr${}_{2}$B${}_{2}$, Fe${}_{2}$B${}_{2}$, and Zr${}_{2}$B${}_{2}$. The $d_{B-B}$, $d_{M-B}$, and *t* parameters are defined in Figure 1. All the values are in Angstroms (Å) and, for comparison purposes, the data in brackets are taken from the literature.

	Cr${}_{2}$B${}_{2}$		Fe${}_{2}$B${}_{2}$		Zr${}_{2}$B${}_{2}$	
	Pmma	P6/mmm	Pmma	P6/mmm	Pmma	P6/mmm
*a*	2.885	2.919	2.823	2.913	3.084	3.159
	(2.860 ^4^) (2.930 ^5^)	(2.921 ^1^) (2.926 ^2^)	(2.800 ^4^) (2.770 ^5^)		(3.07 ^7^)	(3.144 ^1^) (3.160 ^2^) (3.134 ^3^)
*b*	2.870	2.919	2.787	2.913	3.281	3.159
	(2.850 ^4^) (2.870 ^5^)	(2.921 ^1^) (2.926 ^2^)	(2.680 ^4^) (2.820 ^5^)		(3.27 ^7^)	(3.144 ^1^) (3.160 ^2^) (3.134 ^3^)
$d_{B-B}$	1.810	1.685	1.813	1.682	1.721	1.824
		(1.689 ^2^)			(1.71 ^7^)	(1.825 ^2^) (1.891 ^3^)
$d_{M-B}$	2.10	2.15	2.04	2.15	2.46	2.49
	(2.11 ^4^)		(2.05 ^4^)		(2.43 ^7^)	(2.49 ^3^)
*t*	2.122	2.662	2.134	2.680	2.817	3.382
		(2.651 ^1^) (2.639 ^2^)	(2.13 ^6^)		(2.84 ^7^)	(3.391 ^1^) (3.380 ^2^) (3.385 ^3^)

^1^ Ref. [21]. ^2^ Ref. [22]. ^3^ Ref. [23]. ^4^ Ref. [24]. ^5^ Ref. [25]. ^6^ Ref. [26]. ^7^ Ref. [27].

**Table 2 materials-16-05104-t002:** Cohesive energy ($E_{coh}$), highest frequency at the Γ point (ν), and charge transfer from TM to B (Δq) for Cr${}_{2}$B${}_{2}$, Fe${}_{2}$B${}_{2}$, and Zr${}_{2}$B${}_{2}$. The values in bold are to highlight the structure with higher $E_{coh}$.

	Cr2B2		Fe2B2		Zr2B2	
	Pmma	P6/mmm	Pmma	P6/mmm	Pmma	P6/mmm
$E_{coh}$ (eV)	**6.222**	6.201	**6.901**	6.830	8.050	**8.087**
ν (cm${}^{-1}$)	762.45	1003.50	772.74	976.01	928.02	744.12
Δq (*e*)	−0.76	−0.61	−0.37	−0.40	−1.17	−0.86

**Table 3 materials-16-05104-t003:** Total magnetic moment ($\mu_{tot}$), magnetic moment of the TM atoms ($\mu_{\mathrm{TM}}$), magnetic moment induced on the boron atoms ($\mu_{B}$), magnetic ground state (MGS), the total energy difference between the FM and the AFM configurations ($\Delta E_{\mathrm{FM}-\mathrm{AFM}}$), and the critical temperature ($T_{c}$) estimated using Equation (Equation 3) for Cr${}_{2}$B${}_{2}$, Fe${}_{2}$B${}_{2}$, and Zr${}_{2}$B${}_{2}$.

	Cr${}_{2}$B${}_{2}$		Fe${}_{2}$B${}_{2}$		Zr${}_{2}$B${}_{2}$	
	Pmma	** P6/mmm **	Pmma	** P6/mmm **	Pmma	** P6/mmm **
$\mu_{tot}$ ($\mu_{B}$/unit cell)	2.56	0.63	2.69	0.00	0.00	0.00
$\mu_{\mathrm{TM}}$ ($\mu_{B}$/ion)	1.03	0.31	1.26	2.06/−2.06	0.00	0.00
$\mu_{B}$ ($\mu_{B}$/ion)	−0.05	−0.01	−0.05	0.00	0.00	0.00
MGS	FM	FM	FM	AFM	Non-magnetic	Non-magnetic
$\Delta E_{\mathrm{FM}-\mathrm{AFM}}$ (meV/unit cell)	−104.38	−0.29	−108.12	46.84	-	-
$T_{c}$ (K)	403	1	418	181	-	-

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
