# Peer review of "Exploring the Structural, Electronic, Magnetic, and Transport Properties of 2D Cr, Fe, and Zr Monoborides"

_materials, 2023, doi:10.3390/ma16145104_

Round 1

Reviewer 1 Report

attached

I had already mentioned above.

Author Response

1. The generalized gradient approximation (GGA) with the Perdew-Burke-Ernzerhofer (PBE) functional is used for the exchange correlation functional. However, it is well known fact that 3d states of TM ions can not be correctly described by simply GGA. Therefore, I would like to recommend that authors should calculate the properties with GGA+U or more powerful method HSE (hybrid) and compare with GGA results.

RESPONSE: We thank the Reviewer for this remark. We are preparing a separate paper that will include the DFT+U calculations. The "U" parameters for the DFT+U treatment of TM d-electrons will be calculated using the linear response theory. The lattice parameters and atomic positions will be the same as predicted by our GGA (PBE) calculations.

2. Cohesive energy is calculated to check the structural stability of the systems. Along with this, the formation enthalpy/energy with respect to the convex hull rather than the elemental states as discussed in these references [Journal of Physical Chemistry C 118, 19625 (2014), W. Sun, et al., Sci. Adv. 2016, 2 (11), S. Rubab, et al., Phys. Chem. Chem. Phys., 2021, 23, 19472 and Kirklin, S. et al., npj Computational Materials 2015, 1, 15010.] should be present.

RESPONSE: The crystal structures of the studied materials are quite well-established for MBenes which are parent structures of the MAB phases. A more detailed study can be found e.g. in Ref. 3 of our paper (Physical Chemistry Chemical Physics 2022, 24, 11249–11258).

3. Curie temperature should be calculated using Heisenberg Hamiltonian, which is an important parameter for practical realization of the materials as disc-cued in this recent work [Physical Chemistry Chemical Physics 25, 838 (2023)].

RESPONSE: We thank the Reviewer for this remark. We have added critical temperature values estimated within the mean-field approximation in Table 3. And also a new paragraph has been added to the text:
"The energy difference $\Delta E_{\mathrm {FM-AFM }}=E_{\mathrm{FM}}-E_{\mathrm{AFM}}$ can be used to evaluate the exchange interaction $J_{\mathrm{NN}}$ between TM atoms at nearest neighbor (NN) positions. The exchange energy for a system of interacting atomic moments, $\mathbf{S}_i$, can be described by the Heisenberg model
\begin{equation}
E_{tot}=E_0-\frac{1}{2} \sum_{i \neq j} J_{i j} \mathbf{S}_i \cdot \mathbf{S}_j
,\end{equation}
where $E_0$ is the total energy excluding spin-spin interactions and in our case $S_i=S_j=S$. For ferromagnetically or antiferromagnetically coupled TM ions at NN positions $-2 J_{\mathrm{NN}} S^2=\Delta E_{\mathrm {FM-AFM }}$. The critical temperature (Curie or Néel temperature), $T_c$, in the mean field approximation (MFA) can be estimated from
\begin{equation}
T_c \simeq \frac{2}{3}  k_B \cdot \left( J_{\mathrm{NN}} S^2\right) = \frac{1}{3} k_B \cdot \lvert \Delta E_{\mathrm {FM-AFM }} \rvert
\label{tc}
.\end{equation}
The $T_c$ values are collected in Tab.~\ref{table:Magnetic}. As it can be seen from the table, we get a considerably large value of $T_c=418$~K for ortho-Fe$_{2}$B$_{2}$. However, as mentioned above, a more accurate investigation \cite{Ozdemir2021} predicts an AFM ground state with $T_c=115$~K. Interestingly, our calculations predict an AFM ground state for hex-Fe$_{2}$B$_{2}$ with $T_c=181$~K."

4. How about the SOC and the spin-direction, please refer to PHYSICAL REVIEW B 106, L060407 (2022) and add a discussion.

RESPONSE: A SOC and noncolinear magnetism will be included in the follow-up paper that we have mentioned above.

5. Finally, I would like to comments that magnetism on TM ions should be described in terms of different spin states like low, medium, or high spin states.

RESPONSE: We thank the Reviewer for this remark. We do agree with the Reviewer and we will take this advice for the next paper. In the present work, we are using the same description of magnetism as was found in the literature on the topic [e.g. in Phys. Rev. B 103, 144424 (2021)].

Reviewer 2 Report

This manuscript "Exploring structural, electronic, magnetic, and transport properties of 2D Cr, Fe, and Zr monoborides" by Isabel Arias-Camacho and Nevill Gonzalez Szwacki is submitted to Materials journal. The authors performed theoretical calculations for orthorhombic and hexagonal M2B2 MBenes in Density Functional Theory (QE, GGA, PAW setup). The crystal structures were optimized and phonon dispersions are checked. The optimized lattice parameters and interatomic distances are found in agreement with the literature.

This is a comprehensive work with novel interesting results. The manuscript is well organized and written, the conclusions are supported by the results. However, several issues can be revised prior to publication: 

1) The .cif files are expected for any new structures reported. Please include those in Supplementary Information. 

2) Magnetic ordering is found in MGS with AFM order in Fe2B2, it is in agreement with ref. [26 - 10.1103/PhysRevB.103.144424]. This is the main part of the results, so it can be extended. A comparison of the magnetic properties of M2B2 MBenes with the corresponding bulk compounds can be added. 

3) Another minor correction may be to add eV in axis captions in Figs 3 and 4: E-E_Fermi, eV    

I can recommend this manuscript for publishing in Materials after minor corrections.

Author Response

1) The .cif files are expected for any new structures reported. Please include those in Supplementary Information. 

RESPONSE: We thank the Reviewer for this remark. The CIF files will be provided upon request.

2) Magnetic ordering is found in MGS with AFM order in Fe2B2, it is in agreement with ref. [26 - 10.1103/PhysRevB.103.144424]. This is the main part of the results, so it can be extended. A comparison of the magnetic properties of M2B2 MBenes with the corresponding bulk compounds can be added. 

RESPONSE: We thank the Reviewer for this remark. We have added critical temperature values estimated within the mean-field approximation in Table 3. And also a new paragraph has been added to the text:
"The energy difference $\Delta E_{\mathrm {FM-AFM }}=E_{\mathrm{FM}}-E_{\mathrm{AFM}}$ can be used to evaluate the exchange interaction $J_{\mathrm{NN}}$ between TM atoms at nearest neighbor (NN) positions. The exchange energy for a system of interacting atomic moments, $\mathbf{S}_i$, can be described by the Heisenberg model
\begin{equation}
E_{tot}=E_0-\frac{1}{2} \sum_{i \neq j} J_{i j} \mathbf{S}_i \cdot \mathbf{S}_j
,\end{equation}
where $E_0$ is the total energy excluding spin-spin interactions and in our case $S_i=S_j=S$. For ferromagnetically or antiferromagnetically coupled TM ions at NN positions $-2 J_{\mathrm{NN}} S^2=\Delta E_{\mathrm {FM-AFM }}$. The critical temperature (Curie or Néel temperature), $T_c$, in the mean field approximation (MFA) can be estimated from
\begin{equation}
T_c \simeq \frac{2}{3}  k_B \cdot \left( J_{\mathrm{NN}} S^2\right) = \frac{1}{3} k_B \cdot \lvert \Delta E_{\mathrm {FM-AFM }} \rvert
\label{tc}
.\end{equation}
The $T_c$ values are collected in Tab.~\ref{table:Magnetic}. As it can be seen from the table, we get a considerably large value of $T_c=418$~K for ortho-Fe$_{2}$B$_{2}$. However, as mentioned above, a more accurate investigation \cite{Ozdemir2021} predicts an AFM ground state with $T_c=115$~K. Interestingly, our calculations predict an AFM ground state for hex-Fe$_{2}$B$_{2}$ with $T_c=181$~K."

3) Another minor correction may be to add eV in axis captions in Figs 3 and 4: E-E_Fermi, eV.

RESPONSE: Figure 3 was corrected accordingly.

Round 2

Reviewer 1 Report

All the quires has been explain well by the reviewers. Now the manuscript is ready for publications.